# Maladaptive One-Leg Balance Control in Parkinson's Disease

Eric Chevrier [1], Elena Moro [1,2], Pierre Pelissier [1,2], Anna Castrioto [1,2], Paul Krack [3], Valérie Fraix [1,2] and Bettina Debû [2,*]

[1] CHU Grenoble Alpes, Division of Neurology, 38000 Grenoble, France
[2] Grenoble Institute for Neuroscience, Grenoble Alpes University, INSERM, U1216, CHU Grenoble Alpes, 38000 Grenoble, France
[3] Department of Neurology, Center for Parkinson's Disease and Movement Disorders, Bern University Hospital, University of Bern, 3010 Bern, Switzerland
* Correspondence: bettina.debu@univ-grenoble-alpes.fr

**Abstract:** Balance disorders are very frequent in Parkinson's disease (PD). One-leg stance performance is a predictor of fall risk. We investigated one-leg stance strategies in people with PD. We hypothesized that patients would choose, and better perform on, the leg on the least affected body side. Fifty participants with 2 to 19 years of PD duration stood on one leg while ON medication. The leg spontaneously chosen was recorded. Performance was compared between the spontaneously chosen vs. contralateral, and most vs. least stable legs. Influence of disease duration, severity, age, cognition, and motor fluctuations was analyzed. Twenty-eight patients spontaneously stood on the leg of the least affected body side, which was not always the most stable one. The chosen standing leg was influenced by disease duration with a switch between the least vs. most affected body side after seven years of disease duration. Fourteen patients (28%) spontaneously stood on their least stable leg. Thus, some patients with PD choose the least stable leg when asked to perform one-leg stance. It is important to identify these patients since they may be at greater risk of falls and/or gait difficulties. Specific rehabilitation may help prevent such maladaptive strategy.

**Keywords:** asymmetry; balance; Parkinson's disease; rehabilitation

## 1. Introduction

Balance impairment is a common axial sign in advanced Parkinson's disease (PD) that can greatly impact patients' independence and quality of life [1]. While it may not be much performed under static conditions in daily life, one-legged stance reflects the quality of balance control [2]. Several studies [3,4] have shown that the quality of one-leg support is an independent predictor of risk of falling after 60 years of age. Indeed, performance of one-leg stance is measured in numerous scales assessing balance and postural control (e.g., Berg balance test, Tinetti test, Gait and Balance Scale, etcetera) [5].

The spontaneous choice of the support foot for unipodal stance tests is not a random phenomenon [6–10]. It presents very strong agreement (86% in men, 95% in women) with the self-reported leg dominance as assessed using the Waterloo Footedness Questionnaire-Revised [11], and is generally the most stable leg (86% in men, 95% in women) [12]. The spontaneously chosen stance foot is also that on which one stays while initiating gait, stepping on place, or kicking [9]. This functional bias has led to the concept of "postural foot", designating the leg involved in balance control, as opposed to the "motor foot", i.e., the contralateral leg performing the action [8,9,13]. Leg dominance is progressively established during childhood [14,15]. Such functional asymmetry in the coordination of limb movements reflects the asymmetry of brain organization [9,16].

Despite the abundant scientific literature describing axial disorders in advanced PD, the balance strategies developed by patients or their relation to the asymmetrical aspect of the disease are poorly documented. The asymmetric nature of PD pathophysiology leads to

asymmetry in motor signs [17] and may interfere with the leg dominance established since childhood. Accordingly, the most stable side for one-leg balance in people with PD could be determined by an interaction between the initial lateralization and the asymmetrical severity of the disease. In addition, the most stable side may change across the course of the disease, depending on PD-related symptoms' asymmetry, motor fluctuations or other factors. Whatever the outcome of such interaction, one would expect the support foot spontaneously chosen by patients with PD for one-leg stance to be the most effective for maintaining balance, i.e., the most stable. Thus, patients may develop a balance strategies strategy consisting of choosing to stand on the leg least affected by the disease, whether or not this was their initial postural leg. Compensation for balance control asymmetry by increased contribution of the leg of the least affected body side has been observed in response to sinusoidal platform perturbations in patients with asymmetrical disease [18].

To better understand one-leg balance strategies in patients with PD, we investigated their spontaneous choice and balance performance when asked to perform one-leg stance. We hypothesized that patients would choose the leg on the least affected body side, and that standing on that leg would prove more stable than standing on the leg of the most affected body side, i.e., this choice would reflect an adaptive strategy. We also examined whether the balance strategy was consistent across patients with different disease duration.

## 2. Materials and Methods

### 2.1. Subjects

People with PD were consecutively recruited at the Movement Disorders unit of the Centre Hospitalier Universitaire Grenoble Alpes (CHUGA), France. Inclusion criteria were: aged between 30 and 80 years; disease duration $\geq$ 2 years; asymmetrical motor symptoms defined as a two-point difference in the total scores of right and left appendicular "akinesia" and "rigidity" items (3.3–3.8) of the Movement Disorders Society—Unified Parkinson's Disease Rating Scale part III (MDS-UPDRS-III) [19]; dopaminergic medication stable over the three months preceding inclusion. Exclusion criteria were: postural disorders caused by traumatic, rheumatologic, orthopedic or neurological disorder other than PD; deep brain stimulation or other surgeries for PD; cognitive deficit (Mattis Dementia Rating Scale, MDRS, score < 130) [20]; scores >3 on gait (items 3.10) and posture (item 3.13) of the MDS-UPDRS III.

The study was approved by the local ethic committee (IRB 5891), and patients gave written informed consent.

### 2.2. Assessments

Assessments were performed in the early morning, while patients were in their ON medication condition, i.e., about one hour after their usual medication intake. In all patients, PD treatment was optimized, and patients had confirmed that they felt in their "best ON" at the time of the evaluations. Assessments included the MDS-UPDRS parts III and IV, and a one-leg stance test. For the latter, patients were asked to stand on one limb for up to 40 s. Instructions were given orally, without demonstration, in order not to bias the spontaneous choice of standing leg and the chosen leg was recorded. After this first trial, they were asked to stand on the other leg. The duration of stance was recorded.

### 2.3. Primary and Secondary Outcomes

The main outcomes were the leg (least affected vs. most affected body side) spontaneously chosen by the patients, and the duration of one-leg stance. To determine the least affected body side, the sum of the lateral scores for items 3.3 to 3.8 of the MDS-UPDRS-III were computed. The most stable leg was defined as that whereby one-limb stance duration was the longest.

Secondary outcomes were the difference in performance between the most vs. least stable leg, and the relationship between PD duration and the one-leg stance strategy. To that end, the distributions of spontaneously chosen leg (least vs. most affected side) were

plotted against duration of disease to identify possible disease duration cut-off values defining two groups.

*2.4. Statistical Analysis*

Control of the assumptions for parametric statistics showed that performance data were not normally distributed. Therefore, non-parametric statistics were used. Wilcoxon tests were used on the whole sample to compare the duration of one-leg stance on the chosen vs. opposite legs, as well as most vs. least stable legs.

The link between demographical characteristics (age, PD duration, MDS-UPDRS III, MDS-UPDRS IV, medication dosage (using the levodopa equivalent daily dose, LEDD), and MDRS score) and one-limb stance performance was examined using Spearman correlations.

For the secondary outcomes, the Fisher exact test was used to examine the relation between the spontaneously chosen support leg and the most affected body side in the two groups defined based on disease duration distribution. Then, performance and differences in demographical characteristics between the two groups defined on the basis of the leg chosen for the first trial (most affected vs. least affected body side) were examined using non-parametric Mann-Whitney (between groups) and Wilcoxon (within groups) tests.

A $p < 0.05$ was considered significant. Statistical analyses were performed using Statistica (Statistica Advanced 12.6 Dell Software Group) and JASP (JASP Team (2022). JASP (Version 0.15)).

**3. Results**

Fifty patients with PD were included within a 3-year period of inclusion. Their demographic characteristics are presented in Table 1. The most affected body side was the right side in twenty-one patients, while two patients had symmetrical disease at the time of inclusion. None of the patients presented with dyskinesia or dystonia during the evaluation session.

**Table 1.** Demographical characteristics of the study population (N = 50).

| | |
|---|---|
| Age (years) | $57.5 \pm 9.6$ |
| Gender *(M/W)* | 36/14 |
| PD duration (years) | $7.5 \pm 3.6$ |
| MDS-UPDRS III ON Med (/132) | $21.8 \pm 7.8$ |
| MDS-UPDRS IV (/24) | $7.0 \pm 3.9$ |
| MDRS (/144) | $138.8 \pm 3.8$ |
| Medication *(LEDD)* | $1004.1 \pm 424.8$ |
| FOG/Falls *(Number of patients)* | 9/15 |

Unless otherwise specified, data are presented as mean $\pm$ standard deviation; FOG: freezing of gait.

*3.1. Whole Group Analyses*

Twenty-eight patients (56%) spontaneously stood on the leg of the least affected body side, whereas twenty-two patients chose the most affected side. There was no significant difference in stance duration between the two legs (Table 2). However, comparison of stance duration on the most vs. least stable leg revealed a significant difference ($p < 0.01$) (Table 2), suggesting that some patients did not spontaneously chose their most stable leg to stand on. Indeed, 14 patients (28%) spontaneously stood on their least stable leg, whether or not it was the leg on the least affected body side. Twelve patients (24%) were able to maintain balance during 40 s on either leg.

**Table 2.** Disease duration and one-leg stance performance in the two groups defined as a function of leg spontaneously chosen (referred to as preferred leg).

| | Total Sample (N = 50) | Least Affected Body Side (N = 28) | Most Affected Body Side (N = 22) |
|---|---|---|---|
| Disease duration (years) | 7.5 ± 3.6 | 4.7 ± 1.8 | 10.4 ± 2.6 * |
| Stance duration preferred leg (sec) | 26.2 ± 13.9 | 29.9 ± 13.1 | 21.4 ± 13.5 * |
| Stance duration opposite leg (sec) | 25.8 ± 15.3 | 32.1 ± 12.1 | 19.0 ± 16.1 * |
| Stance duration most stable leg (sec) | 29.8 ± 13.6 | 33.2 ± 10.6 | 26.3 ± 15.5 |
| Stance duration least stable leg (sec) | 22.2 ± 14.7 ß | 26.5 ± 13.1 ß | 18.0 ± 15.2 ß |

*: between groups significant difference; ß: within group significant difference.

There was an effect of disease duration on the choice of the leg to stand on (Figure 1, upper panel). Patients who spontaneously stood on the leg of the least affected body side had significantly shorter disease durations ($p < 0.01$). In addition, they outperformed patients who chose the leg on the most affected body side (chosen leg $p < 0.05$; opposite leg $p < 0.01$). There was no difference in disease duration between patients who chose their least vs. most stable leg (Figure 1, lower panel). Notably, ten of the fourteen patients who chose to stand on their least stable leg had freezing of gait (FOG, eight) or falls (four), two having both, compared to ten among the thirty-six other patients (seven with FOG, five with falls, and two having both).

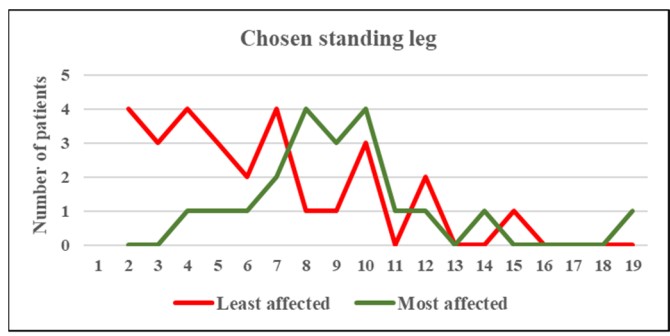

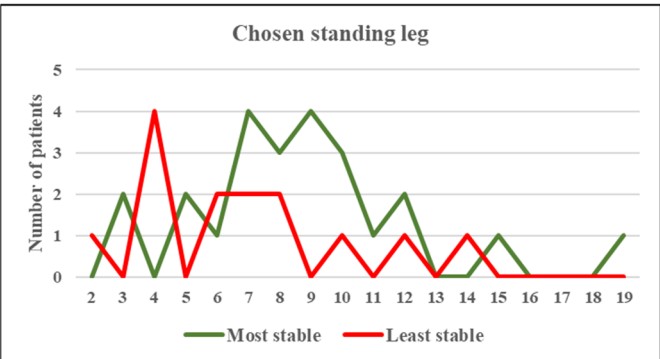

**Figure 1.** Plots of data distribution for spontaneous support leg as a function of disease duration. Upper panel: least affected vs. most affected body side. Lower panel: most stable vs least stable.

In the total sample, correlations were significant between one-leg stance duration and age (spontaneously chosen leg: rho = −0.38, $p < 0.01$; least stable leg: rho = −0.28, $p < 0.05$). In addition, for the least stable leg only, the correlation between one-leg stance duration and disease duration (rho = −0.29, $p = 0.05$) and MDRS score (rho = 0.32, $p < 0.05$) were also significant. There was no significant correlation between one-leg stance performance and MDS-UPDRS IV and MDS-UPDRS III scores or with LEDD. Thus, altogether, one-limb

stance performance decreased with age, and, as disease duration or cognitive dysfunction progressed, it decreased on the least stable leg.

*3.2. Secondary Outcomes*

The distribution of the spontaneously chosen leg (least vs. most affected body side) as a function of disease duration suggested that patients with PD duration up to seven years mostly chose to stand on the leg of the least affected body side, whereas patients with longer disease duration preferentially chose the leg of the most affected body side (Figure 1, upper panel). Therefore, patients were divided into a group with disease duration ≤7-year (G1, 25 patients) and a group with >7-year disease duration (G2, 25 patients). Fisher exact test confirmed that G1 patients spontaneously chose the leg on the least affected body side ($p < 0.01$), whereas G2 patients stood on the leg of the most affected body side ($p < 0.05$). The difference in age between the two groups fell short of significance, with patients of G2 being somewhat older than patients of G1 (62.3 years vs. 57.4 years, $p = 0.051$). LEDD, MDS-UPDRS-III, MDS-UPDRS-IV and MDRS scores did not differ between the two groups.

A comparison of stance duration between the two groups failed to show any significant difference when patients stood on their spontaneously chosen leg or on their most stable one (Table 3). On the contrary, stance duration differed between the two groups when standing on the leg opposite to the spontaneously chosen one, or on the least stable (ps < 0.05). In both cases, stance duration in G1 greater than in G2, in line with the correlational outcomes in the whole sample.

**Table 3.** Disease duration and one-leg stance performance in the two groups defined as a function of disease duration. G1: disease duration ≤7 years; G2, disease duration >7 years.

| | Total Sample (N = 50) | G1 (N = 25) | G2 (N = 25) |
|---|---|---|---|
| Disease duration (years) | 7.5 ± 3.6 | 5.8 ± 3.0 | 9.5 ± 3.2 β |
| UPDRS III score (/132) | 21.8 ± 7.8 | 21.2 ± 7.4 | 22.1 ± 8.4 |
| MDRS score (/144) | 138.8 ± 3.8 | 139.1 ± 4.0 | 138.1 ± 3.5 |
| Stance duration preferred leg (sec) | 26.2 ± 13.9 | 31.4 ± 12.0 | 20.1 ± 13.8 * |
| Stance duration opposite leg (sec) | 25.8 ± 15.3 | 32.6 ± 10.1 | 17.8 ± 16.7 * |

* significant difference between G1 and G2; β: trend for a significant difference between G1 and G2.

Within group comparisons showed that there was no significant difference in stance duration between the spontaneously chosen and opposite legs in either group, but stance duration on the most stable leg was greater than on the least stable leg (G1: $p < 0.001$; G2: $p < 0.001$; Table 3). So, whatever the group, some patients did not choose their most stable leg to stand on.

An additional analysis was run to try and identify demographic factors predicting the choice of the least stable leg to stand on. Logistic regression showed that age and the MDS-UPDRS III total score had a significant predictive value ($p < 0.01$, $p = 0.01$ respectively). Older age and lower MDS-UPDRS III total score increased the risk of standing on the least stable leg.

## 4. Discussion

The aim of this study was to better understand one-leg balance strategies in people with PD. We had hypothesized that patients would choose the leg of the least affected body side to perform the task, and that performance, i.e., stance duration on that leg would be better than performance on the opposite leg. Our results did not support these hypotheses. First, only 28/50 patients (56%) chose to stand on the leg of the least affected body side, and that leg was not the most stable in nine of them (almost one-third of cases). In addition, in 17 out of the 22 patients (77%) who chose to stand on the leg of the most affected body side, the performance on that leg was better than or equal to performance on the opposite

leg. Thus, whether spontaneously standing on the leg of the least or most affected body side, most patients selected their most stable leg (72% of the whole sample, and 63% of the patients with asymmetrical balance performance). Disease duration influenced the patients' choice, with patients with longer duration preferentially choosing the leg on the most affected body side. However, disease duration did not influence performance on the most stable leg.

Remarkably, 14 patients (28% of the sample) opted to stand on their least stable leg, representing 40% of the patients with a difference in performance between the two legs. FOG, falls, or both were very frequent in this sub-group. This maladaptive choice was independent on disease duration.

### 4.1. One-Leg Stance Strategy in Patients with PD

As expected, performance of one-leg stance decreased with age, and with disease duration and cognitive dysfunction [1,21], although these two factors only influenced performance on the least stable leg. More importantly, disease duration influenced one-leg stance strategy. Most patients with disease duration up to seven years chose the leg of the least affected body side, an adaptive choice in only two-thirds of the cases. Conversely, most patients with longer disease evolution chose the leg on the most affected body side, and that leg was also the most efficient for one-leg stance. Indeed, while one-leg stance stability on the least stable leg decreased with disease duration, it was not so for stability on the spontaneously chosen leg. Thus, these results raise the intriguing possibility that patients with PD switch preferential standing-leg over the course of the disease. Early on, they tend to choose the leg on the least affected side, although it may not be the most stable. Such choice may result from the interactions between lifelong motor organization with a dominant postural leg established during childhood and disease development leading to a greater degradation of motor control on one side than on the other. Indeed, patients may face one of two possibilities: (1) a congruence between dominant postural leg and asymmetry of the disease, the latter initially sparing the dominant leg (this would lead to spontaneously standing on the least affected, also the most stable, leg, i.e., an adaptive strategy); (2) a conflict between the dominant postural leg and a disease preferentially affecting that leg. As a result, patients may be led to spontaneously stand on the least affected leg, which, not being their postural dominant leg, is less stable. In other words, the conflict may lead to a maladaptive strategy. As the disease progress, patients may switch to preferentially standing on the leg of the most affected body side, which proves the most stable leg in most instances.

While confirmation of such a switch in preferential standing leg would require longitudinal data, the notion that the leg on the most affected body side may become the more stable with increasing disease duration is supported by several arguments. It could be related to the effects of dopaminergic medication, as the patients were assessed in the ON medication condition. While reports on the effect of levodopa on postural stability have been contradictory [22,23], posturographic recordings have shown that center of pressure displacement is greater with than without medication [24]. In addition, stiffening is a relevant strategy to facilitate balance control, as it minimizes postural sway and decreases the number of degrees of freedom to be controlled [25]. Stiffening the support limb may be easier for the leg that is already stiffer in an asymmetrical disease such as PD. Comparison of the rigidity-akinesia sub-scores of the two legs lends further support to this hypothesis, as the rigidity-akinesia sub-score of the chosen leg was significantly greater than that of the opposite leg in the G2 group ($2.6 \pm 1.4$ vs. $1.8 \pm 1.4$), that preferentially stood on the most affected leg, whereas the opposite was true in G1 ($1.6 \pm 1.5$ vs. $2.9 \pm 1.5$).

Another hypothesis would lay in the development of motor fluctuations after several years of dopaminergic treatment. However, in our specific sample, there were no differences in MDS-UPDRS IV-scores between G1 and G2, and, more importantly, dyskinesia would be expected to more severely affect the most affected body side, leading patients to choose to stand, and be more stable, on the leg of the least affected body side rather than on the

opposite leg. In addition, while the development of dyskinesia is related to dopaminergic medication dosage, there was no difference in LEDD between the two groups, further arguing against an effect of dyskinesia in the outcomes of this study.

Whatever the reason for the switch in preferred standing leg when disease duration increases, if it actually occurs, it would appear particularly relevant as the link between one-leg stance performance and disease duration was only significant for the least efficient leg.

### 4.2. Maladaptive One-Leg Stance Strategy in People with PD

While there were no differences in one-leg stance duration between the chosen and opposite legs in either group of patients, there were significant differences between the most and least stable legs. This may seem contradictory, as most patients' preferred leg was also the most stable. However, in 12 patients (24%), performance was identical on the two legs, whereas the difference in performance between the two legs was greatest in the 14 patients who chose their least stable leg. Thus, the strategy chosen by these patients appears clearly maladaptive, possibly making them more prone to FOG or falls. Indeed, ten out of these 14 patients (71%) had either FOG or falls, against ten out of the 36 patients (28%) who chose their most stable leg. The outcome of the logistic regression revealed that older age and lesser disease severity, as reflected by the MDS-UPDRS part III total score, were predictive of inappropriate choice, whereas disease duration, MDS-UPDS part IV or MDRS scores were not.

Identifying such maladaptive choice, and providing training targeting an appropriate one-leg balance strategy, could be particularly relevant as one-leg stance has been related to risk of falls in individuals over 60 years of age [4] and in people with PD [25]. In addition, one study reported abnormalities resembling FOG in anticipatory postural adjustments preceding automatic stepping response [26]. Finally, gait initiation is slower in patients with PD than in healthy controls, due to difficulties in weight transfer between the swing and stance legs prior to the first step [27].

Systematically testing one-leg stance as disease progresses and providing relevant training might help prevent upcoming gait and balance difficulties. Such training should aim at helping patients more readily use their most efficient postural leg and transfer weight onto it for step initiation, in order to facilitate gait onset, delay the emergence of FOG, and reduce risk of falling. While such effects await confirmation, effective training would be of paramount importance for preventing FOG, which, once installed, is very resistant to medical or surgical interventions [1,27]. Furthermore, a recent meta-analysis concluded that one-leg stance training leads to balance improvement in the healthy population [2].

### 4.3. Influence of Patients' Characteristics on One-Leg Stance

As expected, one-leg stance performance decreased with age, and, to some extent, disease duration and the MDRS score [25,28,29]. A number of studies have shown a link between cognitive function and balance control [28–32]. Such link, however, is not specific to PD and is also seen in healthy controls [33,34]. Remarkably, the MDRS scores did not differ between G1 and G2, while disease duration did, and patients of G2 tended to be older than patients of G1, comforting a general effect of age on one-leg balance. The lack of link between MDRS score and one-leg stance performance when patients stood on their preferred leg further suggests that cognitive function did not explain the effects observed in the present study. Together with the lack of differences in MDRS scores between patients who chose the least vs. most stable leg to stand on, it appears that the relationship between cognitive function and one-leg balance performance reflected a general effect of aging. Of note, however, in our cohort the patients with longest disease duration were candidates for deep brain stimulation, for which cognitive impairment is a contra-indication. Thus, they may not be representative of the larger PD population, and cognitive status should not be overlooked when trying to anticipate on balance issues or design specific training.

*4.4. Limits of the Study*

The present study has some limitations. First, although very large for this type of study, the sample size remains somewhat limited. In addition, although commonly used, the two-point difference between right and left symptoms' severity might be insufficient to consider the disease as truly asymmetrical.

One should also note that the chosen methodology involved a stance test of predefined maximal duration, rather than asking participants to stand as long as possible, leading to ceiling effects in a significant proportion of the participants. Indeed, almost a quarter of the patients reached the maximal duration with both legs. More importantly, while the so-called postural leg is overwhelmingly preferred by individuals when asked to perform a one-leg stance test [11], it is not possible to exclude that some patients could have purposefully decided to perform the test with their least performing leg, although as they were not told that they would have next to stand on the other leg, this appears unlikely.

Finally, the mechanisms underlying lateral differences in balance control and the switch in one-leg stance efficacy remain elusive [6,7,10].

**5. Conclusions**

The results of the present study suggest that the most efficient strategy for one-leg balance control may change with PD progression. Whereas most patients secured optimal one-leg balance control, an important proportion chose their least stable leg to stand on. These patients may be at greater risk of FOG and falls. Altogether, this argues for developing and testing specific balance training for patients with PD to identify maladaptive strategies and provide proper rehabilitation.

**Author Contributions:** E.C.: methodology, data acquisition, statistical analysis, funding acquisition. E.M.: editing, review, supervision. P.P.: data curation, review. A.C.: data acquisition, review. P.K.: methodology, funding acquisition. V.F.: data acquisition, review. B.D.: data curation, statistical analysis, writing original draft, review. All authors have read and agreed to the published version of the manuscript.

**Funding:** The research was funded by the Acting Foundation for Chronic Diseases (AGIR Fund for Chronic Diseases).

**Data Availability Statement:** The data presented in this study are available on request from the corresponding author. The data are not publicly available due to ethical restrictions.

**Conflicts of Interest:** The authors declare no conflict of interest.

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
