# Peer review of "Maladaptive One-Leg Balance Control in Parkinson’s Disease"

_symmetry, doi:10.3390/sym14122511_

Round 1
Reviewer 1 Report
This article investigates and discusses for one-leg stance strategies in patients with PD. One-foot standing is one of the most common examination procedures performed in neurology. The subject matter is simple, but extremely practical and interesting. Parkinson's disease is appropriate for the scope of this journal because, unlike other diseases with parkinsonism, left-right differences in symptoms are important. This article is well written and the logic flows smoothly.
I believe this article is good enough as it is in its present form.
Author Response
We wish to thank the reviewer for their positive assessment of our work.
Reviewer 2 Report
Balance pathophysiology in PD is unexplored fully and balance disorders are an unmet need in patients. The authors have elaborately evaluated the patients of PD with a one leg stance performance strategies , hypothesising that one-leg stance performance is a predictor of fall risk in patients with PD. The results are interesting, in that almost 28% of their patients chose to stand on the least stable leg. These patients (71%) had a tendency for either FOG or falls, as compared (28%) who chose their most stable leg. Results are well illustrated with plots and tables. In the discussion , the maladaptive one leg stance strategy, is discussed elaborately. The limitations are highlighted adequately.
Author Response

(The authors gave the same response as above.)

Reviewer 3 Report
This is a well-written manuscript conveying important findings on fall risk in PD. While the study design can be improved, even at this stage, the research has scientific merit. A few concerns are listed below:
1. Authors state that subjects were evaluated while ON dopaminergic medication. Given the variability in dose and time to peak plasma levels, it is extremely important to control for time (in hours) post-medication. It is likely that one-leg balance performance would vary depending on time post-administration of the dopaminergic drug (i.e., in relation to approximate peak plasma level). This confound is amplified further by likelihood that many patients were on CR-levodopa. Authors should examine time post-administration of the medication on performance and look at CR+ vs CR- subgroup differences, if feasible. If not, this should be addressed as a limitation.
2. It is not clear why nonparametric statistics were used. In what ways did the data violate assumptions for parametric tests to justify this decision?
3. A key outcome of this research was the impaired decision (lack of introspection) in 28% of the PD sample to spontaneously select the more impaired leg to assess balance. While results failed to identify a clinical or cognitive correlate of this decision, it is possible that analysis of MDRS subscale or item scores or a multivariate (logistic regression) analysis might reveal one or more cognitive markers that predict this impairment. This information would be clinically useful.
4. Table 3 should include G1 vs G2 differences on UPDRS and MDRS for completeness.
5. There is an interesting statement in the Discussion (last line on page 8) that suggests stiffening the support limb may be easier in patients with limb rigidity as the limb is already “stiffer”. While speculative in this context, the hypothesis can be easily tested with data collected for this study. It would not be difficult to examine UPDRS (leg or asymmetry) rigidity scores for leg (side) chosen by the subject to perform the balance task. If the hypothesis is correct, the mean rigidity score(s) would be greater in the chosen leg than the opposite leg. This can then be examined further in greater detail.
Minor point:
6. Spell out FOG either in Table 1 (as a footnote) or prior to Table 1 in the text.
Round 2
Reviewer 3 Report
Authors adequately addressed all prior concerns. No remining issues.